# The Role of Galanin during Bacterial Infection in Larval Zebrafish

**DOI:** 10.3390/cells10082011

**Published:** 2021-08-06

**Authors:** Natalia Nowik, Tomasz K. Prajsnar, Anna Przyborowska, Krzysztof Rakus, Waldemar Sienkiewicz, Herman P. Spaink, Piotr Podlasz

**Affiliations:** 1Department of Animal Anatomy, Faculty of Veterinary Medicine, University of Warmia and Mazury, 10-719 Olsztyn, Poland; natalia.nowik@outlook.com (N.N.); anna.jakimiuk@uwm.edu.pl (A.P.); sienio@uwm.edu.pl (W.S.); 2Department of Animal Sciences and Health, Institute of Biology (IBL), Leiden University, 2333 BE Leiden, The Netherlands; tomasz.prajsnar@uj.edu.pl (T.K.P.); h.p.spaink@biology.leidenuniv.nl (H.P.S.); 3Department of Evolutionary Immunology, Institute of Zoology and Biomedical Research, Faculty of Biology, Jagiellonian University, 30-387 Krakow, Poland; krzysztof.rakus@uj.edu.pl; 4Department of Pathophysiology, Forensic Veterinary and Administration, Faculty of Veterinary Medicine, University of Warmia and Mazury, 10-719 Olsztyn, Poland

**Keywords:** zebrafish, immunity, bacterial infection

## Abstract

Galanin is a peptide that is conserved among different species and plays various roles in an organism, although its entire role is not completely understood. For many years, galanin has been linked mainly with the neurotransmission in the nervous system; however, recent reports underline its role in immunity. Zebrafish (*Danio rerio*) is an intensively developing animal model to study infectious diseases. In this study, we used larval zebrafish to determine the role of galanin in bacterial infection. We showed that knockout of galanin in zebrafish leads to a higher bacterial burden and mortality during *Mycobacterium marinum* and *Staphylococcus aureus* infection, whereas administration of a galanin analogue, NAX 5055, improves the ability of fish to control the infection caused by both pathogens. Moreover, the transcriptomics data revealed that a lower number of genes were regulated in response to mycobacterial infection in *gal*−/− mutants compared with their *gal+/+* wild-type counterparts. We also found that galanin deficiency led to significant changes in immune-related pathways, mostly connected with cytokine and chemokine functions. The results show that galanin acts not only as a neurotransmitter but is also involved in immune response to bacterial infections, demonstrating the complexity of the neuroendocrine system and its possible connection with immunity.

## 1. Introduction

Although the connection between neuroendocrine and immune system was a doubtful matter for many years, it has recently been proved that these both systems cooperate and are complementary to each other [1,2]. This link is regulated by a number of neuropeptides, which play different roles in innate immunity, while in turn, immune cells regulate the neural function by cytokines [3,4,5,6].

Galanin has been already known for almost forty years [7]; however, its entire role remains unrevealed. The galanin family includes galanin, the galanin-like peptide (GALP), the galanin-message associated peptide (GMAP) and alarin. Galanin is present mainly in the central nervous system as well as in the peripheral tissues of many different species [8,9,10,11]. It is also involved in many biological processes such as regulation of the circadian rhythm, brain development, pain, food intake, stress, metabolism and inflammation [12]. Moreover, it has been shown to participate in inflammatory diseases such as inflammatory bowel disease and ulcerative colitis [13,14], skin inflammation [15] or even infectious diseases such as tuberculosis [16]. Galanin acts through its receptors (GALRs) that belong to the G-protein-couple receptor (GPCR) superfamily, which in mammals consists of three subfamilies, namely GalR1, GalR2 and GalR3 [17]. In the zebrafish, no GalR3 gene is present, but GalR1 and GalR2 genes are duplicated, which means there are four galanin receptor genes in zebrafish: GalR1a, GalR1b, GalR2a and GalR2b [18]. The expression of galanin receptors is very common in many tissues of adult zebrafish and appears very early in development, even as a maternal transcript (XX) [19,20,21].

Their structure is composed of three intercellular loops (ICLs), an extracellular N-terminus, three intracellular loops, and three extracellular loops (ECLs). The helix eight at the C-termini, which is located parallel to the membrane, is present in all models [17].

Because of its genetic and physiological similarities to mammals, zebrafish (*Danio rerio*) is an extremely useful animal model to study various non-infectious and inflammatory diseases. The embryos are transparent, they develop ex utero and are susceptible to genetic manipulations, such as gene knockout [22]. Its genome has been fully sequenced and shows 70% similarity with the human counterpart [23]. Zebrafish have been extensively used for studying cancer, inflammatory diseases and non-infectious diseases [24,25,26]. Galanin gene is highly conserved in different species, including the zebrafish [11]. The structure of zebrafish galanin gene is identical with the gene present in mammals: It comprises six exons and contains a signal peptide, galanin and GMAP domains. The mature peptide contains 29 amino acids, and the N-terminal amino acids 1–13 (GWTLNSAGYLLGP) are identical in all species including zebrafish from which galanin has been identified, except in tuna where the serine in position 6 has been substituted with an alanine [27]. This strong evolutionary conservation may suggest important and similar roles of galanin in all vertebrates.

*Mycobacterium tuberculosis* (Mtb) is a pathogen that is responsible for the leading deadly infectious disease of humans, of which the cell wall resembles both the Gram-positive and Gram-negative characteristics. This bacterium is estimated to have infected one-third of the world population and to be responsible for nine million new cases of tuberculosis (TB) [28]. The main feature of TB is the development of granulomas that consist of infected macrophages and necrotic cell debris, which form at the site of infection [29]. Granulomas are a hallmark of latent infection that can persist for many years without clinical signs and can be reactivated into the systemic infection again. In turn, *Staphylococcus aureus* is a Gram-positive opportunistic pathogen able to cause infection in multiple tissues of the host. The range of pathologies varies from local cutaneous lesions to more serious infections such as osteomyelitis and life-threatening sepsis [30,31]. Methicillin, or more often, vancomycin-resistant strains, cause difficulties in treatment and become a huge threat to public health worldwide [32].

Many animal models including zebrafish have been used for mycobacterial and staphylococcal research to identify their molecular background as well as the development of new drugs and current treatment support [33]. Zebrafish have been exploited as a model for TB for almost 20 years since it is a natural host for one of the mycobacterial strains, *Mycobacterium marinum,* which shares 85% of its amino acid identity with the human Mtb [34]. In the zebrafish model, *M. marinum* causes a similar systematic disease with the formation of granulomas and the response of the innate immune system, which is completely functional at 28 hpf (hours post fertilization) [35], without the interference of the adaptive immune system, which is developed after 4–6 weeks post fertilization [36]. This implicates the role of the innate immune system in the development of TB and helps to underline its role in the TB pathogenesis [37]. In contrast, zebrafish is not a natural host of *S. aureus*, although a staphylococcal infection model has been successfully established in zebrafish larvae [38].

NAX 5055 (Gal-B2) is a galanin analogue that was chosen for this study due to its bioavailability, metabolic stability, selectivity for galanin receptors and ability to penetrate the blood–brain barrier (BBB) [39]. Other galanin agonists such as galmic and galnon show little receptor subtype specificity, lower affinity for galanin receptors relative to the native peptide and many off-target effects [39,40].

Galanin is conserved among mammals and lower vertebrates, although its role during an infection has not been precisely determined. In this study, we took a genetic and chemical approach to study the role of galanin in the immune response to bacterial infections in the zebrafish larvae. Our results showed that bacterial infection in the absence of endogenous galanin resulted in increased mortalities and bacterial burden, whereas treatment with galanin analog NAX 5055 rescued the hypersusceptibility of the galanin-deficient larvae.

## 2. Materials and Methods

### 2.1. Fish Maintenance

CRISPR-Cas9 technology was used to generate a galanin mutant (*gal−/−*) as described [41]. The mutant has a 10-base-pair deletion in the 3rd exon of the *gal* gene (Ensembl (GRCz11): ENSDARG00000091377), which results in a loss-of-function allele due to a frameshift and premature stop codon. A detailed description of this mutant is in preparation for a separate publication (unpublished data).

The wild-type *gal+/+* and *gal−/−* mutant larvae were maintained as described previously [42]. Embryos were raised in E3 medium (5 mM NaCl. 0.17 mM KCl. 0.33 mM CaCl_2_. 0.33 mM MgSO_4_) at 28.5 °C. During imaging, fish were kept under anesthesia in egg water containing 0.02% buffered 3-aminobenzoic acid ethyl ester (Tricaine, Sigma-Aldrich, St. Louis, MO, USA).

All fish lines are housed both in the fish facility of the Laboratory of Genomics and Transcriptomics, University of Warmia and Mazury in Olsztyn, Poland, which was built according to the local animal welfare standards and in the fish facility of Leiden University compliant with the directives of the local animal welfare committee. Studies performed on early-life-stage zebrafish larvae and euthanasia do not require Ethic Committee permissions. According to the European Directive 2010/63/EU and Polish law regulations O.J. of 2015, item 266.

### 2.2. Bacteria Preparation

*Mycobacterium marinum* strain 20, labelled with mCherry [43], was grown on Difco Middlebrook 7H10 agar (BD and company) supplemented with 10% oleic acid-albumin-dextrose-catalase (OADC, BD and company), 0.5% glycerol and with hygromycin to select for fluorescence expression vectors. A colony of *M. marinum* was resuspended in Difco Middlebrook 7H9 broth (BD and company) supplemented with 10% albumin-dextrose-catalase (ADC, BD and company), 0.05% Tween 80 (Sigma-Aldrich) and 50 µg/mL hygromycin. Bacterial cultures grown overnight at 28.5 °C were washed in PBS, spun down and resuspended to the desired concentration (to obtain 40 CFU for yolk injections and 120 CFU for caudal vein injections) in 2% polyvinylpyrrolidone (PVP40, CalBiochem) in PBS. Phenol red (Sigma-Aldrich) was added to a concentration of 0.085% to visualize the injection process [43].

The *Staphylococcus aureus* SH1000 strain, carrying the pMV158-mCherry plasmid [44], was grown in a brain heart infusion (BHI) broth medium (Oxoid, Basingstoke, UK) supplemented with tetracycline 5 µg mL^−1^ at 37 °C. Bacteria were cultured until OD600 (optical density at 600 nm) reached approximately 1. After centrifugation (4500× *g*, 10 min), the supernatant was discarded, and the pellet was re-suspended in sterile PBS to the desired concentration (to obtain 20 CFU for yolk injections and 1200 CFU for caudal vein injections [38].

### 2.3. Infection

For yolk infection, embryos were collected immediately after single crossing and infected into the yolk between 4–6 hpf. For caudal vein infection, embryos were dechorionated manually at 24 hpf, anaesthetized and injected individually into the caudal vein using glass microcapillary pipettes with 1 nL of bacterial suspension at 28–30 hpf. The same volume of PBS was injected into the mock control [38,43].

### 2.4. Treatment with NAX 5055

The zebrafish embryos were injected twice with galanin analogue NAX 5055 (a kind gift from Drs. Steve White and Grzegorz Bulaj from the University of Utah [39]). The group infected into the yolk received only one dose of NAX 5055, while those infected into the caudal vein were administered twice. The first injection with 1 nL of NAX 5055 in a concentration of 5 μg/g was performed at the first hour postfertilization into the yolk of the embryos, whereas the second injection of the same concentration was performed together with the bacterial delivery into the caudal vein. During the incubation period, the *gal−/−* larvae were kept in the egg water containing 20 µM of NAX 5055 at 28 °C. The *gal+/+* infected larvae and the mock control fish injected with PBS were kept under the same conditions in the egg water as the treated group.

### 2.5. Imaging and Fluorescence Quantification

A Leica fluorescence (MZ16 FA) stereo microscope was used to take images of zebrafish larvae. During imaging, embryos were kept under anesthesia (0.02% Tricaine, (Sigma) in E3). To quantify fluorescence of bacterial burden in individual embryos, the fluorescent images of embryos with custom-made, dedicated pixel quantification software were performed [45].

### 2.6. RNA Isolation and RT-qPCR

RNA was isolated using in QIAzol lysis reagent (Qiagen, Hilden, Germany) for RNA isolation, which was performed using the miRNeasy mini kit (Qiagen), according to the manufacturer’s instructions. Two hundred nanograms of isolated RNA was reverse transcribed using the iScript cDNA Synthesis Kit (Bio-Rad Laboratories B.V., Hercules, CA, USA) according to the manufacturers’ protocols. For the quantification of mRNA expression, qPCR was carried out using iQ SYBR Green Supermix (Bio-Rad Laboratories B.V.). RT-qPCR was performed on a MyiQ Single-Color Real-Time PCR Detection System (Bio-Rad). Cycling conditions were pre-denaturation for 3 min at 95 °C, followed by 40 cycles of denaturation for 15 s at 95 °C, annealing for 30 s at 60 °C and elongation for 30 s at 72 °C. (0.5 °C increments for every 10 s). For every sample, the Ct value was determined from the Ct value of a non-infected control sample, and the fold change of gene expression was calculated and normalized to the expression levels of a reference gene: *peptidylprolyl isomerase Ab* (*ppiab*), since its expression did not change significantly after *M. marinum* infection in the infected larvae [46]. Results were analyzed using the ΔΔCt method [47]. Data shown are mean ± s.e.m. of three independent experiments. The primer sequences are described in the Appendix A.

### 2.7. RNA-Seq Analysis

Zebrafish embryos (*gal+/+* and *gal−/−*) were either infected intravenously at 28 hpf with *M. marinum* or mock-infected with PBS and were divided into four experimental groups: (i) *gal+/+* uninfected, (ii) *gal+/+* infected, (iii) *gal−/−* uninfected and (iv) *gal−/−* infected. At 4 days post infection (dpi) 15 embryos per group were collected, snap-frozen in liquid nitrogen and stored at −80 °C. Total RNA was extracted according to the manufacturer’s instructions, using TRIzol (Life Technologies, Carlsbad, CA, USA). A total of 2 μg of RNA was used to make RNAseq libraries using the Illumina TruSeq RNA Sample Preparation Kit v2 (Illumina, Inc., San Diego, CA, USA). The resulting mRNAseq library was sequenced using an Illumina HiSeq2500 Instrument (Illumina, Inc.) according to the manufacturer’s instructions with a read length of 2 × 50 nucleotides. Data analysis was performed using Genetiles Software (www.genetiles.com, accessed on 30 December 2015, described in [48]), whereas gene ontology and pathways analysis were performed using DAVID Functional Annotation Tool (https://david.ncifcrf.gov/, accessed on 15 May 2016). Paired analysis was performed using DESeq2 comparing each group (*gal+/+* infected vs.*/gal−/−* infected) to the respective control group (*gal+/+* non-infected vs.*/gal−/−* non-infected). Significantly regulated genes were selected by using *p* < 0.05 and |FoldChange| > 1.5 cutoff.

### 2.8. COPAS Analysis

We used COPAS™ XL (Complex Object Parametric Analyzer and Sorter, Union Biometrica, Holliston, MA, USA) to monitor the development of the infection, fluorescence and spreading of the infection. The COPAS™ XL large particle sorter has been designed for the analysis, sorting and dispensing of objects up to 1.5 mm in diameter based on size, optical density and fluorescence intensity. The 561 nm Solid State laser was used for mCherry detection. Zebrafish embryos were measured alive at indicated time points, to determine their bacterial burden with the COPAS XL using the following settings: Photo multiplier tube voltage: 650 V for red and 0 V for yellow, optical density threshold signal 975 mV (COPAS value: 50) and time of flight minimum 320 μs (COPAS value: 800) [48].

### 2.9. Statistical Analysis

Statistical analysis was performed using GraphPad Prism 8 (GraphPad Software, La Jolla, CA, USA). Survival experiments were evaluated using the Kaplan–Meier method. Comparisons between curves were made using the log rank test. Differences in bacterial burden were statistically tested by one-way ANOVA followed by Tukey’s comparison test (multiple group comparisons). For RT-qPCR, statistical significance was estimated by two-tailed *t*-tests on ln(n)-transformed relative induction folds. Significance (*p*-value) is indicated with ns, non-significant; * *p* < 0.05; ** *p* < 0.01; *** *p* < 0.001, **** *p* < 0.0001. Error bars are the mean ± s.e.m.

## 3. Results

### 3.1. Bacterial Infection in Galanin-Deficient Zebrafish Larvae

In order to investigate the role of galanin during bacterial infection, fluorescent *M. marinum* or *S. aureus* was administered into the yolk of approximately 50 zebrafish larvae at the 4–6 hpf. The infection was monitored as long as possible up to 5 dpi for both host survival and bacterial burden, which was calculated using COPAS flow cytometry, fluorescence microscopy and a dedicated pixel count software.

Infection in the yolk progressed rapidly and caused high mortality in the case of both pathogens used. *M. marinum* yolk infection resulted in significantly higher mortality among the *gal−/−* mutants (73%) than in the *gal+/+* wild-type group (35%) after 96 h post infection (hpi) (Figure 1A). *S. aureus* yolk infection resulted in 100% deaths in the *gal−/−* group, compared to the *gal+/+* wild-type infected group that reached mortality of 40% after 20 hpi (Figure 1B). The progression of *M. marinum* infection was determined using the dedicated pixel quantification software after 5 dpi with the bacterial burden being approximately 1.5 times higher in the galanin-deficient group (*gal−/−**)* compared to the wild-type embryos (*gal+/+*) (Figure 1C). The burden of S. aureus infection was measured with the same method after 15 hpi and was three times higher among the mutants (*gal−/−*) than the wild-type larvae (*gal+/+*) (Figure 1D).

Subsequently, we employed the model of systemic infection. Compared with the yolk infection model, zebrafish embryos were more resistant to the infection induced by intravenous infection, which has been observed before [32]. The injections were performed after 28–30 hpf in approximately 30 larvae, followed until 5 dpf (4 dpi) and measured using the same pixel counting software and COPAS flow cytometry. When injected into the circulation, the *gal−/−* mutant infected with *M. marinum* were more vulnerable to the infection than the control group, reaching mortality of 50% compared to 17% in the *gal+/+* wild-type group (Figure 2A). Furthermore, S. aureus infection into circulation showed higher mortality in the *gal−/−* mutants reaching 82%, compared to 48% in the infected *gal+/+* wild-type siblings (Figure 2B). After 4 dpi, the *M. marinum* infection burden was significantly higher in the *gal−/−* mutant larvae than the *gal+/+* wild-type siblings (Figure 2C). Similarly, *S. aureus* bacterial burden was two times higher in the *gal−/−* group compared to the *gal+/+* wild-type larvae (Figure 2D).

Taken together, the obtained data show that galanin deficiency led to increased host mortality and a higher bacterial burden during bacterial infection suggesting a protective effect of galanin in immunity to bacterial pathogens.

### 3.2. NAX 5055 Treatment in the Infected Galanin Mutants

In order to test how the exogenous galanin analog NAX 5055 influences progression of disease in the mutant zebrafish, the *gal−/−* larvae mutant larvae were injected with NAX 5055 and infected. Yolk infection with *M. marinum* resulted in significantly lower mortality that reached 48% in the *gal−/−* treated mutants group compared to the untreated *gal−/−* mutants (73%) (Figure 1A). NAX 5055 treatment also led to slower progression of *S. aureus* infection in the *gal−/−* mutants with 80% mortality after 20 hpi (Figure 1B) while 100% mortality was observed in the *gal−/−* non-treated group. Yolk *M. marinum* infection resulted in a significantly lower bacterial burden in the *gal−/−* NAX 5055-treated group compared to the untreated *gal−/−* mutant larvae, reducing it to the level observed in the *gal+/+* infected group (Figure 1C). The level of *S. aureus* infection in *gal−/−* mutant larvae after NAX 5055 treatment was not significantly changed when compared to the *gal−/−* untreated mutant group (Figure 1D).

Treatment with NAX 5055 resulted in an improvement in the survival of *gal−/−* mutants infected systematically with *M. marinum*, where the level of mortality reached 38% (Figure 2A). NAX 5055 administration was able to reduce the mortality of infected *S. aureus gal−/−* mutants that reached 65% (Figure 2B). The infection burden upon caudal vein infection with *M. marinum* in the *gal−/−* mutant was significantly reduced after NAX 5055 treatment (Figure 2C), to the level observed in *gal+/+* wild-type counterparts. Next, we found that with *S. aureus,* the infection burden was significantly lowered after NAX 5055 treatment compared to the untreated mutant group; however, it did not reach the level of the wild-type siblings (Figure 2D). In addition, we tested the effect of NAX 5055 in wild-type embryos (Appendix A) and found that galanin analogue supplementation leads to a lower bacterial burden of *M. marinum* compared to untreated embryos. Collectively, the results show that treatment with the galanin analogue NAX 5055 is able to improve survival and reduce bacterial burden of infected *gal−/−* individuals and further suggests the host-protective role of galanin in bacterial infection.

### 3.3. Galanin Deficiency and Immune-Related Gene Expression

Having observed the host-detrimental effect of galanin knockout in bacterial infection, we decided to determine how galanin deficiency affects the expression of immune-related genes. We performed a reverse transcription-quantitative PCR (RT-qPCR) for four genes: *irg1l*, *il1b*, *tnfa* and *cxcl8a* (*il8*) in infected *gal+/+*, *gal−/−* and *gal−/−* larvae treated with NAX 5055. In the *gal−/−* group, the expression of *irg1l* was downregulated when compared to the *gal+/+* zebrafish larvae infected with *M. marinum* (Figure 3A). Moreover, in the same group, the expression of pro-inflammatory cytokines *il1b, tnfa* and *cxcl8a* was also downregulated, in comparison to the wild-type *gal+/+* infected zebrafish larvae (Figure 3A). On the other hand, treatment with NAX 5055 lowered the expression of *irgl1l*, as well as *il1b*, *tnfa* and *cxcl8a* in the *gal−/−* larvae (Figure 3A). We also performed the same analysis to determine the levels of abovementioned genes after *S. aureus* infection. The expression of *irgl1l* was significantly downregulated in the *gal−/−* mutant after staphylococcal infection, whereas treatment with NAX 5055 augmented its mRNA level (Figure 3B). The expression levels of *il1b* and *cxcl8a* were also significantly lower in the *gal−/−* mutant, although we did not notice its significant upregulation after the treatment with NAX 5055 (Figure 3B). The *tnfa* expression was slightly lower after the knockout; however, it was not statistically significant compared to *gal+/+* wild-types (Figure 3B).

Collectively, these results suggest that galanin modulates the production of pro-inflammatory cytokines and therefore could be involved in immune processes during mycobacterial and staphylococcal infection.

### 3.4. Transcriptomic Profiling of Galanin-Deficient Zebrafish Larvae Infected with M. marinum

Knowing that galanin plays a host-protective role during mycobacterial infection in larval zebrafish and is involved in the regulation of the tested immune-related genes, we wanted to determine the differences in in gene regulation between *gal+/+* wild-type larvae and knockout *gal−/−* mutants in response to *M. marinum* infection. We analyzed the transcriptomes of control vs. infected larvae in *gal+/+* (Appendix A) and *gal−/−* larvae (Appendix A) using the whole zebrafish larvae at 4 dpi. In *gal+/+* larvae, the infection led to the upregulation of 1578 genes, whereas only 812 genes were upregulated in the *gal−/−* larvae (Figure 4A) revealing that the loss of galanin has a dampening effect on the infection-induced changes in gene expression. In terms of the downregulated genes, the expression of 643 vs. 599 genes were negatively affected by mycobacterial infection in *gal+/+* vs. *gal−/−* larvae, respectively. Interestingly, only 466 and 56 genes were present in the overlap between these two groups for upregulation and downregulation, respectively (Figure 4A). A comparison of changes in the expression levels and fold change between the infected *gal+/+* and *gal−/−* mutant larvae was visualized by heat maps, which show higher upregulation in the *gal+/+* wild-type group than in the *gal−/−* mutant group (Figure 4B) and, importantly, it reveals that many genes were expressed oppositely to each other depending on the presence of galanin (Figure 4B). To study the effect of galanin knockout on the changes in gene expression caused by mycobacterial infection, we plotted the level of gene expression changed by the infection in the *gal−/−* mutants against infection in the *gal+/+* wild-type group for all genes that were significantly regulated by at least one of these conditions. The plot confirms that more genes are regulated in the *gal+/+* group, where most of them are upregulated, whereas in *gal−/−* mutants, most of the genes are downregulated (Figure 4C).

Next, the Gene Ontology (GO) analyses of significantly regulated genes were performed in both wild-type *gal+/+* and *gal−/−* mutant larvae, after *M. marinum* infection. Significantly affected KEGG pathways and GO:biological processes were determined by submission of the significantly regulated zebrafish genes to DAVID bioinformatic tools (https://david.ncifcrf.gov). Based on the results, we classified and compared the regulated genes according to the significant KEGG pathways and biological processes that they are involved in. After taking a closer look into differences between infected *gal+/+* and *gal−/−* larvae, we identified three processes that were common for both groups, namely *Immune response, Inflammatory response* and *Response to bacterium* (Figure 4D and Table 1). In Immune response, 30 genes were upregulated in the *gal+/+,* while only 9 genes were upregulated in *gal−/−* mutant. Moreover, there was only one common gene for both groups. In *Inflammatory response* and *Response to bacterium,* we identified no common genes for both groups. In the case of these processes, there were markedly more upregulated genes in the *gal+/+* group than in the *gal−/−* mutant larvae (Figure 4D and Table 1). A comparison of the two groups showed that the common biological processes in the *gal+/+* group contained more regulated genes than in the *gal−/−* mutant larvae (Table 1) (Figure 4D). We did not identify any biological processes connected to immune response among downregulated genes in both groups (Table 1). Further analysis showed that the group of KEGG pathways involved in immunity and regulated by mycobacterial infection in the *gal+/+* wild-types and in the *gal−/−* mutants were opposite to each other (Table 1). Actually, the analysis showed that two significantly changed pathways, the *Cytokine-cytokine receptor interaction* and the *JAK-STAT signaling pathway,* were upregulated in the *gal+/+* wild-type group but downregulated in the *gal−/−* mutants (Table 1). Significantly changed biological processes connected with immune response were upregulated in both *gal+/+* and *gal−/−* groups; however, more terms were significantly changed in the *gal+/+* group (Table 1). Among the common biological processes we identified, the only common gene was *ccl19a.1* (*chemokine (C-C motif) ligand 19a, tandem duplicate 1*). All the other genes in the three common processes were different (Table 2). Additionally, we analyzed two pathways that were overlapping between *gal+/+* and *gal−/−* mutants, namely the *Cytokine-cytokine receptor interaction* and the *JAK-STAT signaling pathway* (Table 2). In addition, in this case, only a few genes were shared by both groups (Figure 5A,B). In the *Cytokine-cytokine receptor interaction,* we identify *osmr* (*oncostatin M receptor*) and *csf1r* (*colony stimulating factor 1 receptor*), whereas in the *JAK-STAT signaling pathway,* only *osmr* was common for the two groups (Figure 5A,B).

We also checked if the expression of galanin receptors, galr1a, galr1b, galr2a and galr2b, changed in the *gal−/−* mutants before and after *M. marinum* infection, as well as in the *gal+/+* larvae after infection. There is, however, no significant difference in the expression levels between the groups (Appendix A).

Collectively, these data showed that galanin deficiency led to significant changes in the expression of different genes that affects various pathways and processes in an organism, although most of them are downregulated.

## 4. Discussion

Many neuropeptides such as substance P (SP), neuropeptide Y (NPY), the vasoactive intestinal peptide (VIP) and galanin are not only expressed in nervous system, but also in leukocytes [49]. The neuropeptide galanin was first discovered 30 years ago. The galanin family has been shown to be involved in a wide variety of biological and pathological functions. Although galanin is a well-known neuropeptide, there is not much information about its function in the immune response. Interestingly, another member of the family, GMAP, shows anti-microbial activities and is hypothesized to be part of the innate immune system, since it suppresses *Candida albicans* growth and significantly reduces growth in six out of seven Candida strains [50].

In this study, we generated *gal−/−* mutant zebrafish larvae, which also knocks out GMAP expression. The larvae were infected with two bacteria, *M. marinum* or *S. aureus*, which represent pathogens able to develop drug resistance and are therefore alarming in public health problems. It has previously been shown in the murine model that galanin is connected with inflammatory processes in various organs, and it had been reported e.g., that galanin receptor 3 is involved in skin inflammation in mice [51]. Our results indicated that the knockout of galanin expression resulted in a higher bacterial burden and mortality during a systemic caudal vein and yolk infection with both *M. marinum* and *S. aureus*. We also demonstrated that the administration of the exogenous galanin analogue NAX 5055 into galanin-deficient larvae reduced the progression of infection and lowered mortality after caudal vein infection with both pathogens; however, the treatment was barely effective after yolk infection with *S. aureus*, where mortality exceeded 80% after 16 hpi. While *M. marinum* is a natural zebrafish pathogen, *S. aureus* infection is an experimental model to study bacterial burden and immune response. Infection with the pathogen was successfully established previously; however, the injection was performed at 30hpf [38]. During our study, we decided to infect zebrafish embryos with both bacteria at the same time point, which resulted in severe staphylococcal infection, probably due to rapid bacterial proliferation before the onset of cellular immunity in zebrafish embryos. It was not unusual, since yolk infection led to a rapid progress of infection and mortality after *Cronobacter turicensis* [52] or *S. typhimurium* [53] injection. Additionally, NAX 5055 treatment restored the expression of the pro-inflammatory cytokines and general response to the infection.

Based on the findings, we wanted to check the expression of the most important pro-inflammatory cytokines, namely *il1b*, *tnfa* and *cxcl8a*. We demonstrated that *tnfa* was down-regulated in the *gal−/−* mutant, both after *M. marinum* and *S. aureus* infection. TNFα acts as a membrane-bound protein and plays a central role in inflammation and pathogenesis of various diseases [54]. In zebrafish, Tnfa is an already well-known cytokine that is upregulated during *M. marinum* infection, and the reduction of *tnfa* expression leads to an increased bacterial burden and death of macrophages [55]. Interestingly, upregulation of its expression also results in more severe progression of the disease, showing that the right balance in its pro-inflammatory activity is crucial for proper control of the infection [56]. Moreover, Tnfa in zebrafish is responsible for recruiting leukocytes to the site of infection and chemokines production, rather than phagocyte activation [57].

Similar to *tnfa*, we found il1b to be downregulated in the infected *gal−/−* mutant. Interleukin 1 beta (IL1B) is a crucial activator of immune cells, as it is upregulated during *M. tuberculosis* and *M. marinum* infection in the zebrafish and is strictly connected with *tnfa* activity [58]. Il1b is required for a host response to the infection. It is secreted as an active cytokine from its inactive precursor where an inflammatory response occurs. It is conserved between mammals and zebrafish and its expression is upregulated in zebrafish larvae infected with various bacteria such as *S. typhymurium* or *E. tarda* [59,60,61,62]. Cxcl8a is also an important neutrophil activator and is responsible for their recruitment to the infection site [63]. Galanin expression was found on polymorphonuclear neutrophil (PMN) as it increases their sensitivity to the chemokine CXCL8 [61]. As a result, galanin treatment significantly enhanced the response PMNs of human and mice PMN to CXCL8 [64]. We found *cxcl8a* to be significantly downregulated in the *gal−/−* mutant after both *M. marinum* and *S. aureus* infection, compared to the control infected group and significantly upregulated after NAX 5055 treatment in *M. marinum* infected larvae. Moreover, we tested the expression of *irg1l*, an immunoresponsive gene that is a homolog of the same gene known in mice [60], which was also downregulated in the *gal−/−* mutant larvae after infection with both pathogens. Our study shows that galanin acts pro-inflammatory, while its absence downregulates the expression of particular cytokines. However, not much is known about the role and mechanism of galanin in these processes.

Galanin expression has an impact on macrophages and neutrophils [59,60]. Previous research showed that exposure to exogenous galanin affected the cytokine/chemokine expression of macrophages [64]. Galanin was also shown to act as an immunomodulatory peptide, since it can sensitize polymorphonuclear neutrophils towards pro-inflammatory cytokines in humans and mice [65]. Macrophages play a huge role in the development of granuloma. During our research, we did not focus on the immune cells function; however, we noticed that galanin knockout led to higher number of macrophages, which was reduced by NAX 5055 treatment (Appendix A). This observation can be linked to the granuloma formation that was also increased in the *gal−/−* mutants. Interestingly, treatment with NAX5055 in the *gal+/+* wild-type did not have any significant impact on the number of fluorescent macrophages. Infection with *S. aureus* affects both macrophages and neutrophils, the number of which usually increases, except for an overwhelming infection that results in leukopenia and increased mortality [36]. Looking at the progress of *S. aureus* infection in *gal−/−* larvae, it is possible that high mortality is connected to the affected neutrophil function that needs further research. The alteration of chemokine expression and secretion by galanin indicates a role of galanin in immune cell migration, and this function of galanin is not fully studied yet [64].

To understand the molecular background of the galanin knockout, we performed whole-larvae RNA sequencing, both in non-infected fish and after *M. marinum* infection. We compared and divided the larvae into two groups *gal+/+ with gal−/−* both after mycobacterial infection. Significantly changed genes from both infected groups were analyzed using the DAVID functional annotation tool. In the *gal+/+* wild-type group, most of the affected pathways were connected to the immune response and found to be upregulated. These findings in the *gal+/+* wild-type group correlate with previous study by Benard et al. (2016) [63], where at 4 dpi a similar number of genes were changed due to infection (1165 upregulated and 748 downregulated) and those connected to immunity were upregulated. It has also been shown that a large number of affected genes were immune-related, especially at 4 dpi [63]. However, in the *gal−/−* mutant larvae, we found that significantly less immune-related genes were changed after infection; moreover, significantly affected KEGG pathways were downregulated. Two KEGG pathways were shared between the two groups: The *Cytokine-cytokine receptor interaction* and the *JAK-STAT signaling pathway*; however, only two genes (*osmr* and *csf1r*) were common for both groups. A similar finding was observed among common biological processes. Only three of them were shared between the two groups; however, only one gene (*ccl19a.1*) was common. It indicates that galanin knockdown resulted in a markedly different reaction to the pathogen and immune response compared to the control group, even though the same pathways and processes are involved in the infection. In the *gal−/−* mutant larvae, the *MAPK signaling pathway*, cytokines activity and *JAK-STAT pathways* were regulated, which partially correlates with previous findings in a galanin receptor knockout model, where galanin receptor 3 knockout in mice (GAL3-KO) resulted in lower expression of IL-17A, IL-22, IL-23 and TNF-α [51]. Another study showed that galanin modulates cytokine/chemokine expression in macrophages, which depended on the cell line and additional treatment [65]. *JAK-STAT signaling pathway* plays a significant role during bacterial infection. A canonical JAK/STAT pathway has been characterized from studies in cytokine signaling in mammalian cells [66]. In zebrafish, both JAK and STAT proteins have been characterized as well [67]. There are not many reports about galanin association with the JAK/STAT signaling pathway; however, there are studies that show correlation between galanin production and prolactin production via JAK-STAT [68]. In our work, however, JAK/STAT signaling pathways were upregulated in the *gal+/+* control larvae with 13 regulated genes, while they were downregulated in the *gal−/−* mutants with 5 affected genes. Mitogen-activated protein kinases (MAPKs) are involved in relaying extracellular signals to intracellular responses. Several studies suggest that the MAPK pathway affects mycobacterial pathogenesis [69]. It was shown that the intracellular growth of *Mycobacterium avium* in macrophages depends on the extent of MAPK phosphorylation, indicating a role of the pathway in macrophage activation. Additionally, the activation of MAPK is induced by infection with *M. tuberculosis* and is essential for the mycobacterium-induced production of pro-inflammatory cytokines [69]. Our findings showed that compared to the infected *gal+/+* wild-type, where immune-related genes and processes were upregulated, in the *gal−/−* mutants, the situation was opposite. Among biological processes, we determined terms connected to immunity such as *Immune response, Response to bacterium* and *Inflammatory response*; however, most of the GO terms were identified as metabolic processes. In conclusion, galanin knockout attenuates immune response compared to the wild-type siblings revealing the immunostimulatory role of galanin in bacterial infection.

In summary, we explored the role of galanin during bacterial infection in the zebrafish larvae using two pathogens that are commonly used in zebrafish research, *M. marinum* and *S. aureus*. We discovered that galanin expression is crucial for the ability to control the infection by the innate immune system, as well as that galanin knockout leads to immune suppression. Our results correspond to previous findings in murine models, showing complexity of interactions between immune- and neuroendocrine systems. Interestingly, galanin is suggested to have an anti-inflammatory function in some situations and a pro-inflammatory function in others depending on the conditions. The literature is controversial and currently not conclusive. This could be due to the complexity of the metabolic network signaling and interactions between galanin and its receptor. Additional studies are needed to fully understand these processes and improve current therapies against drug resistant bacterial infections.

## Figures and Tables

**Figure 1 cells-10-02011-f001:**
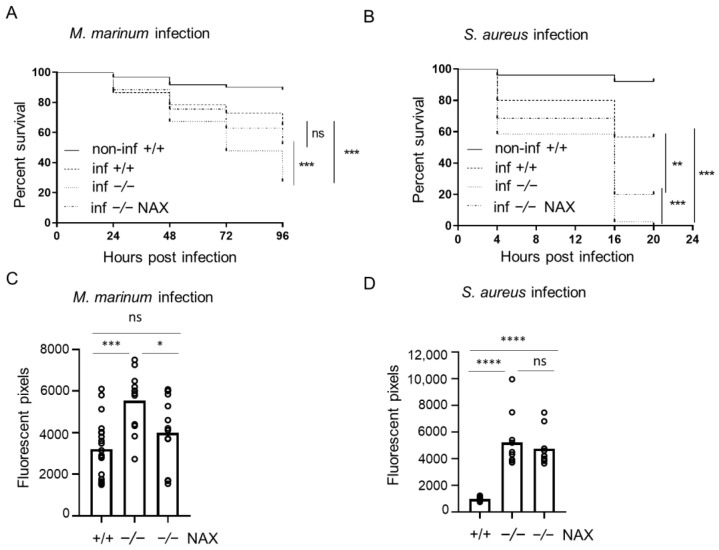
Galanin deficiency results in higher mortality and bacterial burden after yolk infection. (**A**,**B**) Survival rate in mock-injected larvae (non-inf +/+), infected wild-types (inf +/+), infected *gal−/−* mutants (inf -/-) and infected *gal−/−* mutants treated with NAX 5055 (inf -/- NAX) after (**A**) *M. marinum* or (**B**) *S. aureus* yolk infection. (**C**,**D**) Bacterial burden in infected wild-type larvae, infected *gal−/−* mutants and infected *gal−/−* mutants treated with NAX 5055 after (**C**) *M. marinum* or (**D**) *S. aureus* yolk infection. Data are combined from three biological replicates (n = 10 larvae/group). * *p* < 0.05; ** *p* < 0.01; *** *p* < 0.001; **** *p* < 0.0001; ns, non-significant.

**Figure 2 cells-10-02011-f002:**
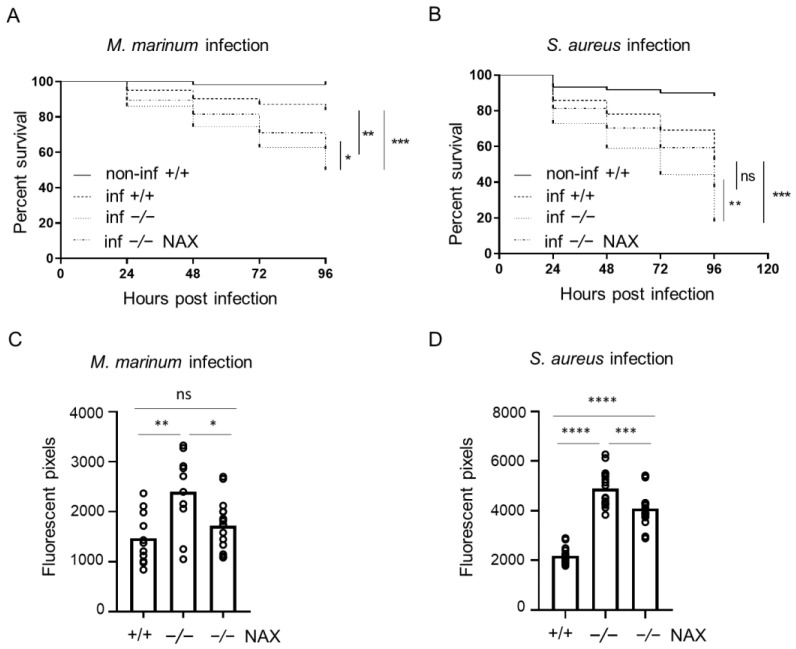
Galanin deficiency results in higher mortality and bacterial burden after vein infection. (**A**,**B**) Survival rate in mock injected larvae (non-inf +/+), infected wild-types (inf +/+), infected *gal−/−* mutants (inf -/-) and infected *gal−/−* mutants treated with NAX 5055 (inf -/- NAX) after (**A**) *M. marinum* or (**B**) *S. aureus* vein infection. (**C**,**D**) Bacterial burden in infected wild-type larvae, infected *gal−/−* mutants and infected *gal−/−* mutants treated with NAX 5055 after (**C**) *M. marinum* or (**D**) *S. aureus* vein infection. Data are combined from three biological replicates (n = 10 larvae/group). * *p* < 0.05; ** *p* < 0.01; *** *p* < 0.001; **** *p* < 0.0001; ns, non-significant.

**Figure 3 cells-10-02011-f003:**
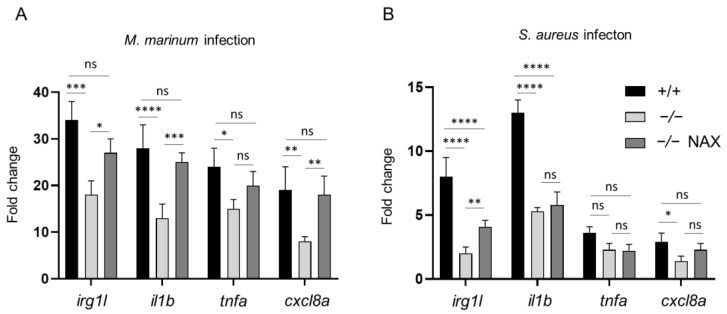
Expression levels of immune-related genes determined by RT-qPCR. *irg1l*, *il1b*, *tnfa* and *cxcl8a* levels after (**A**) *M. marinum* or (**B**) *S. aureus* infection. Data are mean ± s.e.m. of three independent experiments. * *p* < 0.05; ** *p* < 0.01; *** *p* < 0.001; **** *p* < 0.0001 (determined using ANOVA with a Fisher’s LSD post hoc). ns, non-significant.

**Figure 4 cells-10-02011-f004:**
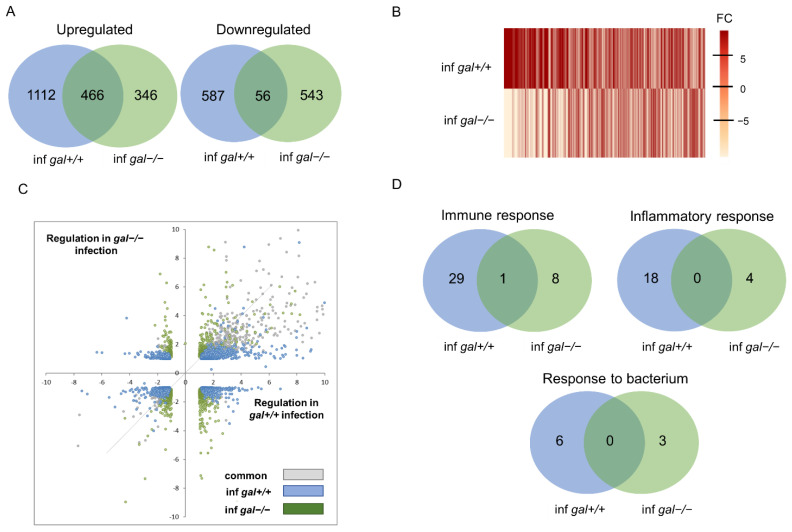
Transcriptome analysis by RNA-seq showing modulation of infection-induced gene regulation by galanin mutation. (**A**) Venn diagram showing overlaps between clusters of genes significantly upregulated or downregulated by infection of *gal+/+* and *gal−/−* mutant larvae; (**B**) heat map displaying the normalized fold changes of transcripts that reached statistical significance as differentially expressed in infected *gal+/+* and infected *gal−/−* zebrafish larvae; (**C**) scatter plot showing the effect of galanin deficiency on gene expression. For all genes showing significant regulation upon infection (blue and gray dots) or the combined infection in the mutant (gray dots). Gray dots represent the overlap between control infection and mutant infection. The gray line indicates the point at which galanin knockout treatment does not alter infection-induced gene regulation; (**D**) GO analysis for upregulated genes in biological processes. Paired analysis was per-formed using DESeq2 by comparing each group to each non-infected group. Significantly regulated genes were selected by using a *p*.adj < 0.05 and |FoldChange| > 1.5 cutoff.

**Figure 5 cells-10-02011-f005:**
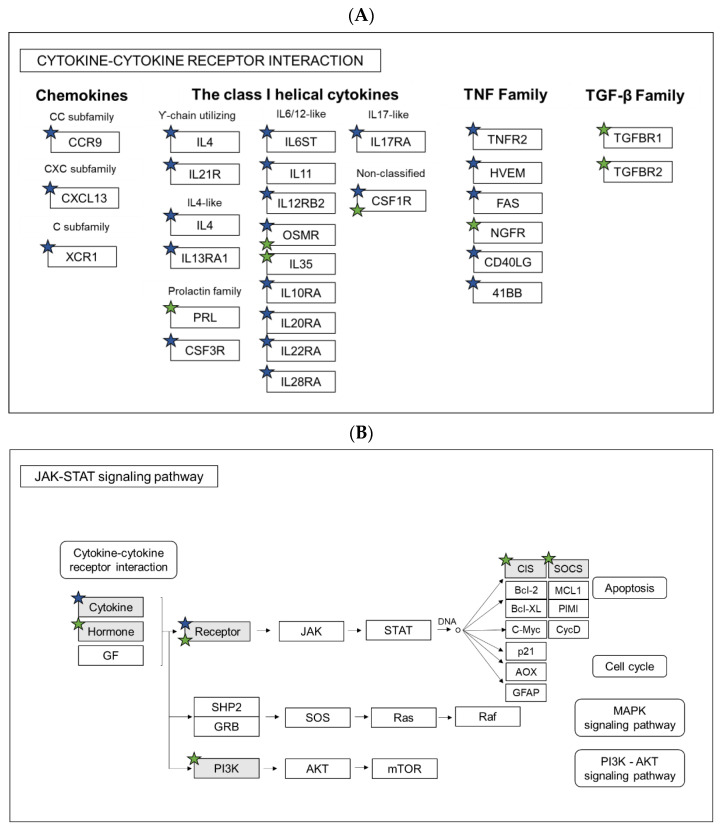
Pathway analysis. Comparison of regulated genes between *gal+/+* and *gal−/−* larvae after *M. marinum* infection in KEGG Pathways: (**A**) *Cytokine-cytokine receptor interaction* and (**B**) *JAK-STAT signaling pathway*. Blue stars indicate genes regulated in *gal+/+,* while green in *gal−/−*.

**Table 1 cells-10-02011-t001:** Comparisons of affected pathways and biological processes in *gal+/+* wild-type and *gal−/−* mutant larvae after *M. marinum* infection. Common processes are marked as grey.

KEGG-Pathways			
**UPREGULATED**			
***gal+/+***		***gal−/−***	
**TERM**	**COUNT**	**TERM**	**COUNT**
*Cytokine-cytokine receptor interaction*	18	*Protein processing in endoplasmic reticulum*	11
*Jak-STAT signaling pathway*	13		
*Herpes simplex infection*	12		
*Cell adhesion molecules (CAMs)*	11		
*FoxO signaling pathway*	11		
*p53 signaling pathway*	10		
*Toll-like receptor signaling pathway*	10		
*Tight junction*	8		
*Insulin resistance*	8		
*Apoptosis*	7		
*Adipocytokine signaling pathway*	7		
*RIG-I-like receptor signaling pathway*	6		
*Cytosolic DNA-sensing pathway*	5		
*Arachidonic acid metabolism*	5		
*Steroid biosynthesis*	4		
**DOWNREGULATED**			
*Cell cycle*	12	*MAPK signaling pathway*	10
*Purine metabolism*	11	*Vascular smooth muscle contraction*	7
*Focal adhesion*	11	*Melanogenesis*	6
*ECM-receptor interaction*	10	*Cytokine-cytokine receptor interaction*	6
*Melanogenesis*	8	*Jak-STAT signaling pathway*	5
*DNA replication*	7	*Glycolysis/Gluconeogenesis*	4
*Pyrimidine metabolism*	6	*Steroid biosynthesis*	3
*Tyrosine metabolism*	5	*Galactose metabolism*	3
*Phototransduction*	4		
*Retinol metabolism*	4		
*Glutathione metabolism*	4		
*Caffeine metabolism*	2		
GO: Biological processes			
***UPREGULATED***			
***gal+/+***		***gal−/−***	
**TERM**	**COUNT**	**TERM**	**COUNT**
*Proteolysis*	46	*Immune response*	9
*Immune response*	30	*Cell redox homeostasis*	5
*Oxidation-reduction process*	27	*Protein folding*	4
*Inflammatory response*	18	*Inflammatory response*	4
*Regulation of cell proliferation*	15	*Response to endoplasmic reticulum stress*	3
*Cell adhesion*	15	*Gene silencing by RNA*	3
*Regulation of apoptotic process*	14	*Polyadenylation-dependent snorna 3′-end processing*	3
*Protein ubiquitination*	12	*CUT catabolic process*	3
*Innate immune response*	11	*Response to bacterium*	3
*Chemotaxis*	8	*Nuclear polyadenylation-dependent rrna catabolic process*	2
*Response to lipopolysaccharide*	8	*DNA methylation involved in gamete generation*	2
*Blood coagulation*	7	*Pirna metabolic process*	2
*Cytokine-mediated signaling pathway*	7	*Exonucleolytic nuclear-transcribed mrna catabolic process involved in deadenylation-dependent decay*	2
*Transmembrane receptor protein tyrosine kinase signaling pathway*	7	*U4 snrna 3′-end processing*	2
*Cell-matrix adhesion*	6	*Nuclear-transcribed mrna catabolic process, exonucleolytic, 3′-5′*	2
*Response to bacterium*	6	*Exonucleolytic trimming to generate mature 3′-end of 5.8S rrna from tricistronic rrna transcript (SSU-rrna, 5.8S rrna, LSU-rrna)*	2
*Peptidyl-tyrosine autophosphorylation*	6	*Nuclear retention of pre-mrna with aberrant 3′-ends at the site of transcription*	2
*Cell chemotaxis*	6	*Glycogen metabolic process*	2
*Positive regulation of apoptotic process*	6	*Adrenal gland development*	2
*Positive regulation of MAPK cascade*	5	*Nuclear polyadenylation-dependent trna catabolic process*	2
*Regulation of cell growth*	5		
*Epiboly involved in gastrulation with mouth forming second*	5		
*Immune system process*	4		
*Response to wounding*	4		
*Circadian rhythm*	4		
*Neutrophil chemotaxis*	4		
*Regulation of cell migration*	4		
*Definitive hemopoiesis*	4		
*Positive regulation of hematopoietic progenitor cell differentiation*	3		
*Activation of MAPK activity*	3		
*Regulation of hematopoietic progenitor cell differentiation*	3		
*Response to glucose*	3		
*Response to cytokine*	3		
*Intracellular sequestering of iron ion*	3		
*Response to heat*	3		
*Iron ion transport*	3		
*Peptide cross-linking*	3		
*Regulation of inflammatory response*	3		
*Response to cadmium ion*	3		
*Macrophage differentiation*	3		
*Response to mechanical stimulus*	3		
*Activation of innate immune response*	2		
*Regulation of cysteine-type endopeptidase activity involved in apoptotic process*	2		
*Plasminogen activation*	2		
**DOWNREGULATED**			
*Transport*	35	*Regulation of transcription, DNA-templated*	32
*Transmembrane transport*	15	*Transport*	30
*Visual perception*	14	*Oxidation-reduction process*	18
*Response to stimulus*	13	*Transmembrane transport*	17
*DNA replication*	10	*Regulation of cell growth*	6
*Cellular response to light stimulus*	8	*Steroid hormone mediated signaling pathway*	6
*Phototransduction*	8	*Potassium ion transmembrane transport*	5
*Protein-chromophore linkage*	7	*Single organismal cell-cell adhesion*	4
*DNA replication initiation*	6	*Neurotransmitter transport*	4
*Retina development in camera-type eye*	6	*One-carbon metabolic process*	4
*Melanosome organization*	5	*Heart contraction*	4
*Microtubule-based process*	5	*Muscle contraction*	3
*Developmental pigmentation*	5	*Carbohydrate transport*	3
*Erythrocyte differentiation*	4	*Glucose 6-phosphate metabolic process*	2
*Embryonic hemopoiesis*	4	*Sodium-dependent phosphate transport*	2
*Oxygen transport*	4	*Mitotic G1 DNA damage checkpoint*	2
*Mitotic cell cycle*	4		
*Nucleobase-containing compound metabolic process*	4		
*Positive regulation of cell proliferation*	4		
*Spindle assembly*	3		
*Skeletal muscle contraction*	3		
*Response to light stimulus*	3		
*Error-free translation synthesis*	2		
*DNA strand elongation*	2		
*Regulation of hematopoietic stem cell differentiation*	2		
*Melanin biosynthetic process*	2		
*Detection of chemical stimulus involved in sensory perception of bitter taste*	2		
*Negative regulation of cysteine-type endopeptidase activity*	2		
*Regulation of G2/M transition of mitotic cell cycle*	2		

**Table 2 cells-10-02011-t002:** Comparisons of affected genes in biological processes in *gal+/+* wild-type and *gal−/−* mutant larvae after *M. marinum* infection. Common genes are marked as grey.

Immune Response	
***gal+/+***	***gal−/−***
*CD74 molecule, major histocompatibility complex, class II invariant chain a(cd74a)*	*Fas ligand (TNF superfamily, member 6)(faslg)*
*CX chemokine ligand 34b, duplicate 11(cxl34b.11)*	*chemokine (C-C motif) ligand 19a, tandem duplicate 2(ccl19a.2)*
*Fas cell surface death receptor(fas)*	*chemokine CCL-C17a(LOC100002392)*
*chemokine (C-C motif) ligand 19a, tandem duplicate 1(ccl19a.1)*	*complement component 7a(c7a)*
*chemokine (C-C motif) ligand 36, duplicate 1(ccl36.1)*	*interleukin 10(il10)*
*chemokine (C-X-C motif) ligand 11, duplicate 1(cxcl11.1)*	*si:rp71-1i20.2(si:rp71-1i20.2)*
*chemokine (C-X-C motif) ligand 11, duplicate 7(cxcl11.7)*	*tumor necrosis factor (ligand) superfamily, member 10(tnfsf10)*
*chemokine (C-X-C motif) ligand 19(cxcl19)*	*uncharacterized LOC101882211(LOC101882211)*
*chemokine (C-X-C motif) ligand 8b, duplicate 1(cxcl8b.1)*	*zmp:0000000652(zmp:0000000652)*
*complement factor properdin(cfp)*	
*interleukin 4(il4)*	
*lymphocyte cytosolic protein 2a(lcp2a)*	
*negative regulator of reactive oxygen species(nrros)*	
*nuclear factor, interleukin 3 regulated(nfil3)*	
*proteoglycan 4b(prg4b)*	
*serpin peptidase inhibitor, clade E (nexin, plasminogen activator inhibitor type 1), member 1(serpine1)*	
*si:ch211-137i24.12(si:ch211-137i24.12)*	
*si:ch73-27e22.6(si:ch73-27e22.6)*	
*si:ch73-27e22.7(si:ch73-27e22.7)*	
*si:ch73-44m9.1(si:ch73-44m9.1)*	
*si:dkey-19a16.4(si:dkey-19a16.4)*	
*si:dkey-253d23.4(si:dkey-253d23.4)*	
*si:rp71-36a1.1(si:rp71-36a1.1)*	
*titin-like(LOC101883412)*	
*tumor necrosis factor (ligand) superfamily, member 11(tnfsf11)*	
*tumor necrosis factor receptor superfamily, member 1B(tnfrsf1b)*	
*tumor necrosis factor, alpha-induced protein 3(tnfaip3)*	
*uncharacterized LOC101883645(LOC101883645)*	
*uncharacterized LOC101885444(LOC101885444)*	
*zgc:153759(zgc:153759)*	
**Inflammatory Response**	
*C5a anaphylatoxin chemotactic receptor(c5ar1)*	*chemokine (C-C motif) ligand 19a, tandem duplicate 2(ccl19a.2)*
*CD40 molecule, TNF receptor superfamily member 5(cd40)*	*interleukin 10(il10)*
*E74-like factor 3 (ets domain transcription factor, epithelial-specific) (elf3)*	*toll-like receptor 22(tlr22)*
*Fas cell surface death receptor(fas)*	*toll-like receptor 5b(tlr5b)*
*chemokine (C-C motif) ligand 19a, tandem duplicate 1(ccl19a.1)*	
*chemokine (C-X-C motif) ligand 11, duplicate 1(cxcl11.1)*	
*chemokine (C-X-C motif) ligand 11, duplicate 7(cxcl11.7)*	
*chemokine (C-X-C motif) ligand 19(cxcl19)*	
*colony stimulating factor 1 receptor, a(csf1ra)*	
*myeloid differentiation primary response 88(myd88)*	
*negative regulator of reactive oxygen species(nrros)*	
*nitric oxide synthase 2a, inducible(nos2a)*	
*prostaglandin E receptor 2b (subtype EP2)(ptger2b)*	
*serum/glucocorticoid regulated kinase 1(sgk1)*	
*toll-like receptor 5a(tlr5a)*	
*tumor necrosis factor receptor superfamily, member 1B(tnfrsf1b)*	
*v-rel avian reticuloendotheliosis viral oncogene homolog(rel)*	
*zgc:153759(zgc:153759)*	
**Response to Bacterium**	
*CCAAT/enhancer binding protein (C/EBP), beta(cebpb)*	*matrix metallopeptidase 9(mmp9)*
*coagulation factor V(f5)*	*toll-like receptor 22(tlr22)*
*hepcidin antimicrobial peptide(hamp)*	*transferrin-a(tfa)*
*immunoresponsive gene 1, like(irg1l)*	
*interleukin-1 receptor-associated kinase 4(irak4)*	
*leukocyte cell-derived chemotaxin 2 like(lect2l)*	

## Data Availability

The raw data presented in this study are available as non-published materials.

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
