# Peer review of "The Role of Galanin during Bacterial Infection in Larval Zebrafish"

_cells, 2021, doi:10.3390/cells10082011_

Round 1
Reviewer 1 Report
Nowik et al reported that galanin played an important role in controlling bacterial infections in zebrafish. Zebrafish larva survival was significantly reduced when galanin was knockout. This reduced survival can be rescued by galanin analogue NAX 5055. The authors also investigated the mechanisms behind galanin mediated response via measuring the expression levels of 4 immune genes and transcriptomic analyses. I only have one question and one comment.
- Why NAX5055 only partially rescued S. aureus infections in gal-/- mutants? In Figure 1B, NAX treated group was still significantly different from inf +/+. In Figure 1D, -/- NAX was at the same level as untreated -/-. Whilst this study focused on Mycobacterium marinum, the results of S. aureus infections should also be discussed.
- It was difficult to see the contents in Figure 5, particularly 5A. The fonts are simply too small to read.
Author Response
Dear reviewer,
Thank you for your comments and suggestions. We find them all relevant and valuable. We hope that we answered the questions in an expected way and that all unclarities were logically explained. The answers can be found below.
Kind regards,
Authors
REVIEWER 1
Nowik et al reported that galanin played an important role in controlling bacterial infections in zebrafish. Zebrafish larva survival was significantly reduced when galanin was knockout. This reduced survival can be rescued by galanin analogue NAX 5055. The authors also investigated the mechanisms behind galanin mediated response via measuring the expression levels of 4 immune genes and transcriptomic analyses. I only have one question and one comment.
1. Why NAX5055 only partially rescued S. aureus infections in gal-/- mutants? In Figure 1B, NAX treated group was still significantly different from inf +/+. In Figure 1D, -/- NAX was at the same level as untreated -/-. Whilst this study focused on Mycobacterium marinum, the results of S. aureus infections should also be discussed.
We believe that the differences are mainly caused by the route of infection. Yolk sac injection site at this time point (4 – 6 hpf) was presumably not the most suitable for this pathogen (probably due to rapid proliferation before the onset of cellular immunity in zebrafish embryos).
S. aureus missing discussion part has been added between lines 424 – 437.
2. It was difficult to see the contents in Figure 5, particularly 5A. The fonts are simply too small to read.
The figure has been modified to be more readable.

Reviewer 2 Report
In the current study from Nowik and colleagues, the effects of loss of galanin in infection models of zebrafish were analyzed. The study design is trustworthy and the conclusion drawn are supported by the data. However, some issues remain:
Minor issues:
- Ad more info about GAL in the intro
- Describe the rationale why you took NAX 5055 and not another GAL agonist or GAL itself.
- Figure labeling, Subfigures big letters - figure legends small letters.
- Figure 5: unreadable, too small, too crowded.
- Discussion: citation 43, NO! GAL3 loss led to less severe skin inflammation! Please cite carefully!
- Spelling Discussion line 388, 422 e.g.: re-sponsible.
Major issues:
- Please show which GAL-receptors are expressed (qRT-PCR data, etc..).
- Please add the following controls (s.aureus infection as effects are more pronounced): I) infected +/+ + NAX 5055 II) infected +/+ + GAL
- Critically discuss the effects of your CRISPR approach on GMAP as it is cleaved from the pre-pro peptide!
- Critically discuss Fig.1D NAX shows only minor effects on s.aureus numbers, your statement in line 269 is too strong!
- You see interesting effects on cxcl8a mRNA, please discuss this also in the context of GAL and put it into context with literature on GAL´s effects on IL-8 responsive cells (e.g. PMN)
- There are no immune cell infiltration data sets, please at least discuss which immune cell population is key and will be affected, better: show stainings.
Author Response
Dear reviewer,
Thank you for your comments and suggestions. We found them valuable and we will take them into consideration in the future studies. However, we were not able to perform any additional experiments regarding the comments, we went through them and applied in the manuscript. We hope that we sufficiently answered to Your comments and corrected all typos. Below You can find our answers.
Kind regards,
Authors
REVIEWER 2
In the current study from Nowik and colleagues, the effects of loss of galanin in infection models of zebrafish were analyzed. The study design is trustworthy and the conclusion drawn are supported by the data. However, some issues remain:
Minor issues:
Ad more info about GAL in the intro
More information about GAL has been added between lines 39-65.
Describe the rationale why you took NAX 5055 and not another GAL agonist or GAL itself.
Clarification has been added between lines 92-96.
Figure labeling, Subfigures big letters - figure legends small letters.
The labeling has been adjusted.
Figure 5: unreadable, too small, too crowded.
The figure has been modified to be more readable.
Discussion: citation 43, NO! GAL3 loss led to less severe skin inflammation! Please cite carefully!
The citation has been adjusted in line 420-421.
Spelling Discussion line 388, 422 e.g.: re-sponsible.
The mistakes have been corrected.
Major issues:
Please show which GAL-receptors are expressed (qRT-PCR data, etc..).
We have unfortunately not checked how the receptors are expressed in the gal-/- mutants by qRT-PCR method. Due to the lack of time we are not able to study this topic in a short time period. Instead we can show expression of the receptors based on RNAseq results, however the expression level is not significantly changed upon gal knockout. We added the results as non-publishable data.
Please add the following controls (S.aureus infection as effects are more pronounced): I) infected +/+ + NAX 5055 II) infected +/+ + GAL
We performed the analysis only for M. marinum infection, which we added to the data set as additional figure (Fig.S1). We have not done the same analysis for S. aureus.
Critically discuss the effects of your CRISPR approach on GMAP as it is cleaved from the pre-pro peptide!
This information has been added between lines 412 – 417.
Critically discuss Fig.1D NAX shows only minor effects on S.aureus numbers, your statement in line 269 is too strong!
The statement in line 269 has been changed. S. aureus missing discussion part has been added between lines 424 – 437.
You see interesting effects on cxcl8a mRNA, please discuss this also in the context of GAL and put it into context with literature on GAL´s effects on IL-8 responsive cells (e.g. PMN).
The additional discussion about cxcl8a has been added between lines 467-470.
There are no immune cell infiltration data sets, please at least discuss which immune cell population is key and will be affected, better: show stainings.
We discussed the key immune cells between lines 479-495. We added one supplementary figure to the dataset, however we did not study this topic into details.
Round 2
Reviewer 2 Report
I thank the authors for answering nearly all questions raised and critically discussing their findings. The quality of the manuscript raised significantly.
Some minor points remain:
We have unfortunately not checked how the receptors are expressed in the gal-/- mutants by qRT-PCR method. Due to the lack of time we are not able to study this topic in a short time period. Instead we can show expression of the receptors based on RNAseq results, however the expression level is not significantly changed upon gal knockout. We added the results as non-publishable data.
Where is the data added? Please provide in text that you validated known expression patterns non-quantitatively in your model system and refer to the datasets.
Lane 64 Typo
The additional discussion about cxcl8a has been added between lines 467-470
This should be citation 61!? What do two species mean? Write human/mouse instead, the sentence is incomplete.
We discussed the key immune cells between lines 479-495. We added one supplementary figure to the dataset, however, we did not study this topic in details.
Fine with me, please provide in the figure legend what the figure is about. The granuloma discussion is viable and adds interesting insights into the possible “chemokine-like” function.
Author Response
Dear reviewer,
Thank you again for your comments and questions. We hope that we sufficiently answered to Your comments and corrected all typos. Below You can find our answers.
Kind regards,
Authors
I thank the authors for answering nearly all questions raised and critically discussing their findings.
The quality of the manuscript raised significantly.
Some minor points remain:
1. We have unfortunately not checked how the receptors are expressed in the gal-/- mutants by qRT-PCR method. Due to the lack of time we are not able to study this topic in a short time period. Instead we can show expression of the receptors based on RNAseq results, however the expression level is
not significantly changed upon gal knockout. We added the results as non-publishable data.
Where is the data added? Please provide in text that you validated known expression patterns non-quantitatively in your model system and refer to the datasets.
A supplementary table (Table S2) has been added to the manuscript and mentioned in lines 399-402.
We also added some more information about galanin receptors in zebrafish in the Introduction (lines 46-55). We realize that it would be beneficial to validate the expression patterns in the mutant larvae and after the infection, but we hope that the provided information is acceptable.
2. Lane 64 Typo
The mistake has been corrected.
3. The additional discussion about cxcl8a has been added between lines 467-470
This should be citation 61!? What do two species mean? Write human/mouse instead, the sentence is incomplete.
This was indeed our mistake with the citation. It has been changed to 61. The sentence has been corrected.
4. We discussed the key immune cells between lines 479-495. We added one supplementary figure to the dataset, however, we did not study this topic in details.
Fine with me, please provide in the figure legend what the figure is about. The granuloma discussion is viable and adds interesting insights into the possible “chemokine-like” function.
The description has been added to the figure’s legend.
